# Development of a Quality Index to Evaluate the Impact of Abiotic Stress in Saline Soils in the Geothermal Zone of Los Negritos, Michoacán, Mexico

**Yanely Bahena-Osorio** [1], **Marina Olivia Franco-Hernández** [2], **José J. Pueyo** [3,*] and **María Soledad Vásquez-Murrieta** [1,*]

1   Escuela Nacional de Ciencias Biológicas, Instituto Politécnico Nacional, Prolongación Carpio y Plan de Ayala s/n, Col. Santo Tomás, Del. Miguel Hidalgo, Ciudad de México C.P. 11340, Mexico; yan.bhna@gmail.com

2   Unidad Profesional Interdisciplinaria de Biotecnología, Instituto Politécnico Nacional, Av. Acueducto, Barrio la Laguna Ticomán, Ciudad de México C.P. 07340, Mexico; mofrancoh@hotmail.com

3   Institute of Agricultural Sciences, ICA-CSIC, Serrano 115-bis, E-28006 Madrid, Spain

*   Correspondence: jj.pueyo@csic.es (J.J.P.); murrieta@hotmail.com (M.S.V.-M.); Tel.: +34-69-6474-599 (J.J.P.); +52-55-5729-6000 (ext. 62352) (M.S.V.-M.)

**Abstract:** In recent years, salinity-induced soil quality impairment and the misuse of management practices have led to the reduced productivity of agroecosystems. This has prompted a search for simple and effective agricultural management strategies that improve the sustainability of agricultural production through soil quality assessments. In this context, the objective of this study was to establish an integrated soil quality index (SQI) by assessing the influence of different types of abiotic stress in two different seasons, using physical, chemical and biological indicators at three sites in the geothermal zone of "Los Negritos", Michoacán, Mexico. Thirty-nine indicators related to soil fertility attributes and C, N, P, and S cycling—identified as the total dataset (TDS)—were evaluated. Principal component analysis (PCA) and the Spearman correlation matrix ($r^2 \geq 0.6$) were used to calculate the SQI using an integrated quality index (IQI) equation, with the indicators total nitrogen (TN), cation exchange capacity (CEC), lithium (Li), and zinc (Zn) identified as the minimum dataset (MDS). Significantly higher SQI values related to the better performance of soil functions were detected during the rainy season.

**Keywords:** salinity; abiotic stress; soil quality indicators; soil properties; minimum dataset; principal component analysis; integrated quality index

## 1. Introduction

The United Nations has estimated that the world population will reach 8.5 billion people in 2030, 9.7 billion in 2050, and 11.2 billion in 2100, posing a challenge to agricultural production in the face of the global threat of soil degradation from high salt concentrations [1–3]. Salinity is defined as the accumulation of water-soluble mineral salts in the soil, with either primary (natural processes) or secondary (human-induced) causes. It is measured based on the electrical conductivity of the soil saturation extract (ECe, dS/m) and—depending on the level—impacts agricultural production, environmental health, and consequently socioeconomic conditions [4,5]. The most recent data issued by the Food and Agriculture Organization of the United Nations (FAO) indicates that 118 countries contain salinity-affected soils, comprising an estimated 424 million hectares in the upper soil layer (0–30 cm) and 833 million hectares of subsoil (30–100 cm) [6]. Therefore, a transition to the proper management of land use is necessary to better understand the role and make decisions that promote sustainable agriculture [7,8].

Soil assessments that allow for monitoring quality for a specific purpose are usually carried out using physical, chemical, and biological indicators that demonstrate an ability

to perform a particular function. It is recommended that such assessments should meet—as far as is practicable—universal criteria for different conditions and soil types; represent the precise function for the purpose they were developed; elucidate ecosystem processes; be easily measurable, reliable, integrative, and sensitive to soil alterations; discern between normal situations and stress situations, either by soil management or by climatic conditions on different scales and/or time periods; and be measurable in terms of time and cost [8–13].

The reliability of establishing a soil quality index (SQI) depends on using the appropriate analytical methods and integration based on the score of the information of the evaluated indicators; the main evaluation methods are mathematical and statistical in nature. The process seeks to obtain a minimum dataset (MDS) that adequately represents the total dataset (TDS) on quality and that contributes to reducing the cost of evaluation. The factorial analysis usually involves (1) the selection of a TDS of the soil properties in relation to a specific function focused on within the objective of the study, (2) the choice and interpretation of an MDS, and (3) the integration of the scores in an index [8,14,15].

Until now, there has been no consistent methodology for selecting a universal dataset to characterize soil quality across regions and scales, and it has been proposed that the establishment of an SQI be conducted according to specific purposes [14,16]. The main disadvantages are the unequivocal interpretation or the lack of reference values, which affect the subjectivity of the evaluated indicators, which is why it is important to clearly define the objectives of the study [12].

However, the Integrated Quality Index (IQI) is the most widely used index because it has proven to be a flexible, effective, and easy quantification tool for assessing the quality of a given soil or region. Also, it reduces measurement costs by reducing the number of indicators used, and it avoids collinearity [12,14,16]. Some studies have assessed soil quality using MDS to calculate the IQI. For example, Yuan et al. [8] assessed 12 soil properties and established the SQI using parameters such as soil organic carbon (SOC), microbial biomass carbon (MBC), total potassium (TK), oxidation-reduction potential (Eh), and Mn (II) in soils with aquaculture activities. Mamehpour et al. [14] determined 24 variables and, as a result, EC, OC, SAR, CEC, bioavailable Fe, and total Cd and Pb were selected as MDS to evaluate soils in semi-arid calcareous ecosystems. Liu et al. [16], based on 26 parameters, established an MDS with soil organic matter (SOM), total nitrogen (TN), pH, dehydrogenase, and arbuscular mycorrhiza for IQI in agricultural soils.

The objective of this study was to establish a quality index under different soil management practices, integrating the effects of different levels of salinity, as well as temporality, to obtain a minimum set of data that represents greater inference on the performance of the soil, and that serves as a quick tool for quantifying the quality of these soils. Here, we measured different parameters that we used to define an IQI for salinity-affected soils in the Geothermal Zone of Los Negritos, Michoacán, Mexico.

## 2. Materials and Methods

### 2.1. Description of Site and Soil Collection

The site and soil collection were described by Guevara-Luna et al. [17] as a geothermal zone at the boundary of the Trans-Mexican Volcanic Belt; hydrothermal activity has been reported in this area, and is associated with the presence of mud volcanoes with temperatures between 48 and 94 °C at the surface reported. Soil samples (Figure 1) were collected from nine points per plot by sampling the 15–25 cm top layer after removing the top 0–15 cm layer from two arable sites, S1 (20°03′24.432″ N 102°36′36.632″ W) and S2 (20°03′02.817″ N 102°37′37.013″ W), and one non-cultivable site, S3 (20°03′44.75″ N 102°36′46.78″ W), in two seasons (March 2019 and September 2020), coincident with two seasonal variations of an annual cycle (the dry and rainy season, respectively). In the first season, S1 was ready for cultivation, i.e., medium deep furrows were made in the soil using agricultural machinery (tillage) to initiate an agricultural cycle; on the other hand, S2 had a few plant residues of sugar cane (*Saccharum officinarum* L.) and weed growth on the soil surface, i.e., no tillage had been carried out and no initial agricultural cycle was planned. S3 had no history of cultivation due

to its high salinity. In the second season, S1 had a developing maize (*Zea mays* L.) crop, S2 had abundant plant residues due to maize (*Zea mays* L.) harvesting, and S3 remained uncultivated. The soil collected from each site was properly transported (labeled in sterile polyethylene bags) to the laboratory and stored until analyzed for its physical, chemical, and biological attributes related to soil fertility, carbon, nitrogen, and phosphorus cycles.

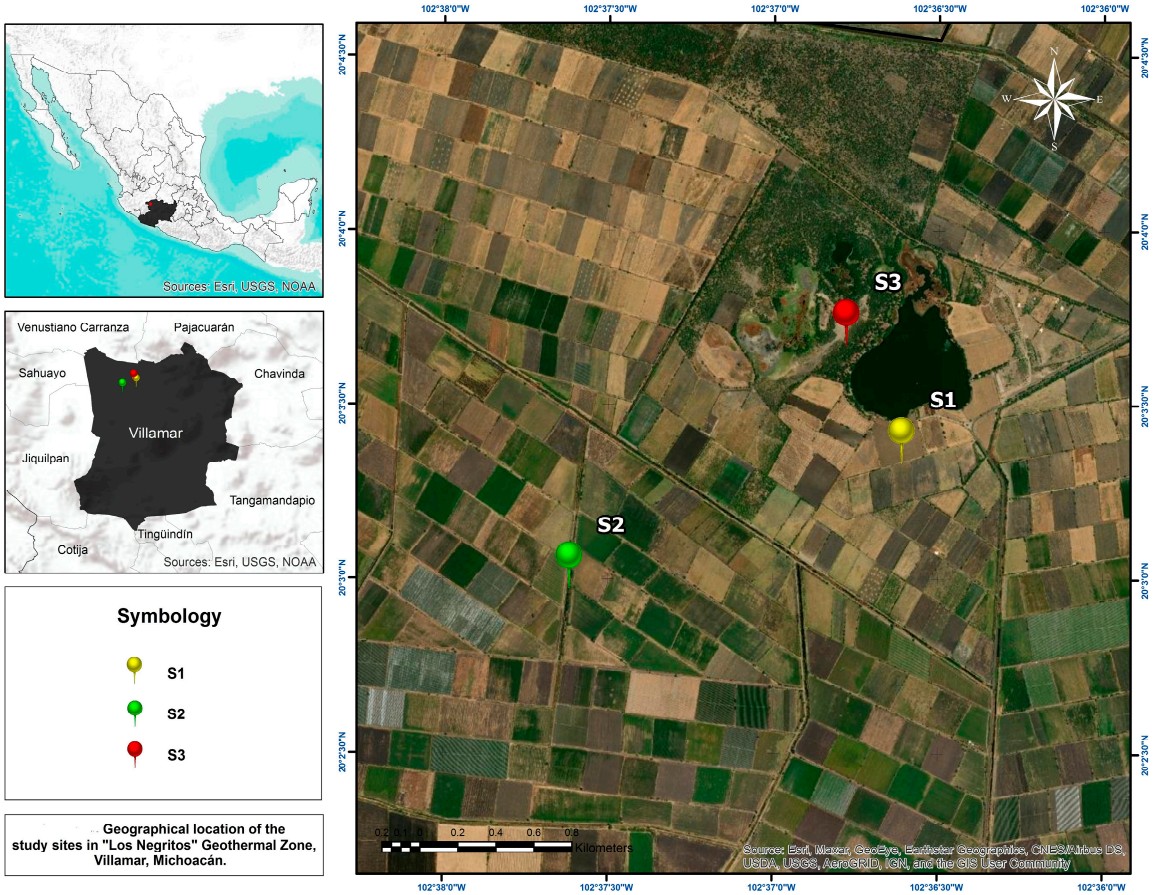

**Figure 1.** Sites from which saline soils were sampled in the geothermal zone of "Los Negritos", Villamar, Michoacán, Mexico.

### 2.2. Soil Quality Indicators

Soil samples were dried at environment temperature and passed through a 2 mm sieve to reduce particle size and remove crop residues. The physical and chemical indicators of relative humidity, water holding capacity (WHC), hydrogen potential (pH), electrical conductivity (EC), cation exchange capacity (CEC), total organic carbon (TOC), total nitrogen (NT) ammonium ($NH_4^+$), nitrate ($NO_3^-$), nitrite ($NO_2^-$) soluble phosphorus ($PO_4^{3-}$), carbonate ($CO_3^{2-}$), bicarbonate ($HCO_3^-$), sulfate ($SO_4^{2-}$), chloride ($Cl^-$), and textural classification were determined as described by Guevara-Luna et al. [17]. Trace elements and major cation concentrations were quantified using acid digestion with $HNO_3$/HCl and inductively coupled plasma optical emission spectroscopy analysis (ICP-OES PerkinElmer Avio 500). Calibration was performed with deionized water and appropriate standards at 1 mg $L^{-1}$ [18]. The biological indicators identified were urease, alkaline phosphatase and acid phosphatase, β-D-glucosidase, and arylsulfatase enzymatic activities, and were identified using modifications of previously established techniques [19–23].

### 2.3. Development of the Soil Quality Index

To determine the SQI, the methodology proposed by Andrews et al. [24], Mamehpour et al. [14], and Li et al. [25] was followed, with the general approach of choosing

the MDS from the TDS of plausible indicators to assess soil quality using multivariate statistical techniques [24,26].

A total of 39 indicators consisting of chemical, physical, and biological properties representing the fertility conditions and the cycle of nutrients C, N, P, and S were evaluated in two seasons at these sites. We performed a two-way variance analysis (ANOVA) of the 39 indicators, with the indicators that showed a significant difference ($p \leq 0.05$) between the analyzed sites selected to be part of the TDS. To identify potential soil indicators for the MDS, a principal component analysis (PCA) was performed on the previously standardized TDS matrix. For each principal component (PC), variables with eigenvalue $\geq 1$ that explained at least 5% of the TDS variation and up to 85% of the cumulative variation within each PC were considered [26].

Subsequently, for each selected PC, each variable was assigned a weight or factor-loading representing the contribution of that variable to the PC composition. Only highly weighted variables from each PC were considered as candidates for the MDS (those that represented absolute values within 10% of the highest factor loading, or $\geq 0.40$). When more than one variable qualified under the same PC, multivariate correlation coefficients (Spearman ($r^2 > 0.6$)) were used to determine whether variables could be considered redundant, and thus were candidates for removal from the MDS.

The indicators considered were those that were highly weighted and non-redundant; however, if the group of variables was correlated, the absolute values of the correlation coefficients of each were summed and it was assumed that the variable with the highest correlation sum best represented the group and formed the MDS. The choice of correlated variables could also be based on practicality of cost, sampling, interpretation, and importance to the study [26].

After defining the MDS, each variable datum was transformed using three types of nonlinear scoring function. "More is better" and "less is better" score curves were applied to indicators when a soil indicator was considered good for soil quality in increasing order (more is better), such as organic carbon, or in decreasing order (less is better), such as salt content, as well as "optimal" scores considering thresholds and reference values of soil properties [27–29]. The first nonlinear scores of the variables were performed using a sigmoidal type of function; Equation (1):

$$\text{SNL} = \frac{1}{1 + \left(\frac{X}{Xm}\right)^b} \tag{1}$$

where SNL is the nonlinear score of the soil indicator, *a* is the maximum score achieved by the function—which is equal to 1 in this study—*X* is the value of the selected soil indicator, *Xm* is the average value of each soil indicator, and *b* is the slope of the equation and is set as $-2.5$ for a "more is better" and 2.5 for a "less is better" curve.

The third score is the threshold value—those soil indicators where the score is equal to 1 when the value is at an optimal level or is equal to 0 when it is at an unacceptable level; Equation (2):

$$\frac{1}{\left[1 + ((B - L)/(x - L))^{2S(B + x - 2L)}\right]}, \tag{2}$$

where *B* is the reference value of the soil indicator where the score is equal to 0.5, *L* is the lower threshold, *S* is the slope of the tangent to the curve at the base line, and *x* is the value of the soil indicator. Threshold and base line values were based on the literature, reference data, expert opinion, or previously observed measured values in ideal soil conditions for the specific purpose concerned.

After calculating the scores, the SQI described by Doran and Parkin [30] was established with the following equation; Equation (3):

$$\text{SQI}_w = \sum_{i=1}^{n} Wi \times Si \tag{3}$$

where SQI$_w$ is the soil quality index (weighted additive), *Si* is the MDS indicator score, *n* is the number of soil indicators in the MDS, and *Wi* is the weighting value of the soil indicators, determined by the variation of each respective PC (%) standardized to the unit.

### 2.4. Soil Quality Grades

Once the quality index had been obtained, to establish different levels the interval of the index obtained (maximum minus the minimum) was divided by the desired number of classifications. The result was used as the base for each level, adding that value to the lowest value of the index to obtain the upper limit of the first interval, and so on, until the upper range was reached [31,32].

### 2.5. Statistical Analysis

The evaluation of all the physical, chemical, and biological indicators was carried out by replicates (nine repetitions) at separate times. Data distribution was based on the Shapiro–Wilk test, with a significance level of $p \leq 0.05$. Significant differences between site indicators in both seasons were determined with a significance level of $p \leq 0.05$ using ANOVA. To demonstrate the correlations between the variables, a Spearman correlation matrix was developed ($r^2 > 0.6$). Statistical analyses (ANOVA, PCA, Spearman correlation) were performed with MINITAB 17 and R 4.21 (www.r-project.org, accessed on 10 November 2022).

## 3. Results and Discussion

### 3.1. Soil Quality Indicators

Based on the evaluation of 39 soil parameters and a two-way ANOVA, a significant difference was observed in most of the estimated parameters between soils from the same season and between seasons; for example, the nutrient content TOC, TN, NH$_4^+$, NO$_3^-$, and PO$_4^{3-}$ (Table 1). The soils were classified as sandy clay loam and sandy loam; however, although they showed the same textural class, their sand, silt, and clay contents differed significantly. It has been shown that the balance of particles forming the structure of a soil influences water movement, aeration, and the ease of root growth [16]. The area is characterized by soils classified as having light to extreme salinity, with EC values ranging from 1.18 to 34.38 dS m$^{-1}$ [33].

**Table 1.** Values of physical, chemical, and biological indicators of analyzed soils from "Los Negritos" geothermal zone in Villamar, Michoacán, Mexico.

| Site | | S1 | S2 | S3 | S1 | S2 | S3 |
|---|---|---|---|---|---|---|---|
| Indicators | Unit | Dry Season | | | Rainy Season | | |
| Moisture content | % | 7.30 ± 1.11 Bb | 11.21 ± 1.80 aB | 10.11 ± 2.45 aB | 22.72 ± 1.29 bA | 24.65 ± 1.43 aA | 16.05 ± 0.97 cA |
| WHC | mg kg$^{-1}$ | 925.7 ± 116.2 bA | 1204.9 ± 63.7 aA | 991.1 ± 84.4 bA | 919.17 ± 15.48 bA | 1010.1 ± 56.8 aB | 864.50 ± 21.07 cB |
| pH | | 6.63 ± 0.29 cB | 6.96 ± 0.12 bB | 9.12 ± 0.14 aB | 7.74 ± 0.09 bB | 7.61 ± 0.13 cA | 9.31 ± 0.02 aA |
| EC | dS m$^{-1}$ at 25 °C | 2.23 ± 0.41 cA | 12.41 ± 1.86 bA | 34.38 ± 2.77 aA | 1.18 ± 0.53 cB | 10.2 ± 2.26 bB | 26.97 ± 4.32 aB |
| CEC | cmolc kg$^{-1}$ | 7.01 ± 4.30 aB | 5.06 ± 2.34 aB | 1.31 ± 0.65 bB | 43.33 ± 2.72 bA | 54.58 ± 6.09 aA | 31.81 ± 3.25 cA |
| TOC | | 149.33 ± 4.45 aB | 126.93 ± 4.56 bB | 28.80 ± 5.37 cB | 527.84 ± 15.29 aA | 586.4 ± 184.9 aA | 496.48 ± 17.48 aA |
| TN | | 1.63 ± 0.17 bA | 2.07 ± 0.29 aA | 0.049 ± 0.38 cB | 1.74 ± 0.18 aA | 1.76 ± 0.12 aB | 0.213 ± 0.06 bA |
| NH$_4^+$ | | 32.21 ± 13.90 bA | 85.01 ± 38.7 aA | 3.33 ± 1.04 cB | 15.99 ± 0.76 bB | 18.97 ± 1.98 aB | 5.72 ± 1.15 cA |
| NO$_2^-$ | | 95.98 ± 6.91 aA | 91.31 ± 1.98 aA | 92.59 ± 1.16 aA | 66.91 ± 5.11 bB | 78.46 ± 2.44 aB | 68.02 ± 3.93 bB |
| NO$_3^-$ | | 1908 ± 947 aA | 55.23 ± 7.94 bB | 98.42 ± 25.62 bB | 94.8 ± 33.0 bB | 287.6 ± 46.6 aA | 349.5 ± 132.3 aA |
| PO$_4^{3-}$ | | 103.8 ± 42.5 bA | 218.2 ± 40.3 aA | 76.28 ± 14.56 bA | 34.40 ± 5.58 bB | 33.18 ± 2.27 bB | 61.95 ± 6.32 aB |
| CO$_3^{2-}$ | mg kg$^{-1}$ | ND | ND | 186.70 ± 17.71 B | ND | ND | 319.4 ± 11.7 A |
| HCO$_3^-$ | | 111.85 ± 13.21 bB | 130.49 ± 8.04 aB | 32.20 ± 17.79 cB | 230.5 ± 40.7 bA | 325.4 ± 30.5 aA | 122.0 ± 68.2 cA |
| SO$_4^{2-}$ | | 723.9 ± 61.0 cA | 1372.7 ± 236.7 bA | 1917.4 ± 116.4 aA | 303.3 ± 142.4 cB | 429.81 ± 15.61 bB | 1832 ± 74.6 aA |
| Cl$^-$ | | 110.93 ± 22.90 cA | 205.95 ± 8.02 bA | 450.19 ± 3.65 aA | 20.25 ± 0.0 cB | 225.02 ± 26.2 bA | 475.9 ± 88.9 aA |
| Sand | | 516.31 ± 25.0 bB | 536.31 ± 10.0 aB | 542.97 ± 10.0 aB | 593.48 ± 10.0 aA | 596.8 ± 596.8 aA | 583.48 ± 25.0 aA |
| Clay | | 326.95 ± 5.0 aA | 43.62 ± 5.00 cB | 260.3 ± 30.0 bA | 106.59 ± 10.0 bB | 63.19 ± 10.0 cA | 243.19 ± 10.0 aA |
| Silt | | 156.74 ± 27.84 cB | 420.07 ± 8.66 aA | 196.74 ± 20.0 bA | 299.93 ± 17.32 bA | 340.0 ± 39.7 aB | 173.33 ± 21.79 cA |

**Table 1.** *Cont.*

| Site | | S1 | S2 | S3 | S1 | S2 | S3 |
|---|---|---|---|---|---|---|---|
| Indicators | Unit | Dry Season | | | Rainy Season | | |
| As | | 118.39 ± 67.0 aA | 61.9 ± 44.1 abA | 20.7 ± 32.4 bA | 157.1 ± 45.3 aA | 19.2 ± 57.5 bA | 17.7 ± 35.5 bA |
| Ca | | 10198 ± 6438 bA | 14381 ± 3989 bA | 47741 ± 24224 aA | 18638 ± 15578 bA | 11605 ± 1461 bA | 49495 ± 17203 aA |
| Cd | | 7.96 ± 2.70 abA | 7.52 ± 1.65 bA | 13.79 ± 8.36 aA | 10.12 ± 7.29 aA | 6.21 ± 0.62 abB | 2.84 ± 3.37 bB |
| Co | | 8.64 ± 4.35 abA | 11.47 ± 1.02 aA | 7.49 ± 2.46 bA | 10.79 ± 1.30 aA | 11.60 ± 1.31 aA | 3.97 ± 4.93 bA |
| Cr | | 51.32 ± 28.41 aA | 70.52 ± 10.40 aA | 55.03 ± 5.76 aA | 70.82 ± 6.81 aA | 77.93 ± 11.67 aA | 27.6 ± 33.3 bB |
| Cu | | 31.94 ± 17.34 bA | 45.26 ± 3.50 aA | 37.45 ± 8.06 abA | 42.91 ± 1.48 aA | 43.71 ± 9.32 aA | 17.27 ± 20.70 bB |
| Fe | | 14213 ± 8124 bA | 21353 ± 1951 aA | 9431 ± 4811 bA | 18173 ± 3697 aA | 22844 ± 3210 aA | 11027 ± 5014 bA |
| Li | | 45.21 ± 15.85 bB | 57.11 ± 3.85 abA | 73.09 ± 26.18 aA | 61.50 ± 7.57 aA | 60.53 ± 6.18 aA | 70.70 ± 40.1 aA |
| Mg | mg kg$^{-1}$ | 6879 ± 3910 bA | 10290 ± 980 bA | 19113 ± 8582 aA | 11497 ± 5583 bA | 11316 ± 1607 bA | 23635 ± 5384 aA |
| Mn | | 353.0 ± 200.9 aA | 388.5 ± 91.8 aA | 309.7 ± 117.2 aA | 457.9 ± 44.5 aA | 333.3 ± 72.5 bA | 267.6 ± 51.8 bA |
| Mo | | 15.34 ± 6.82 aB | 12.5 ± 5.32 aA | 4.84 ± 5.88 bA | 44.02 ± 36.6 aA | 38.0 ± 48.1 abA | 1.59 ± 4.77 bA |
| Ni | | 36.6 ± 33.8 aA | 44.86 ± 22.77 aA | 22.18 ± 5.27 aA | 28.99 ± 3.61 aA | 30.96 ± 7.05 aA | 12.32 ± 14.88 bA |
| Sr | | 84.2 ± 48.8 bA | 147.99 ± 24.62 bA | 521 ± 271 aA | 206.7 ± 177.8 bA | 159.83 ± 25.16 bA | 549.7 ± 171.9 aA |
| Ti | | 465.0 ± 264.8 bA | 974.6 ± 191.9 aA | 360.2 ± 211.3 bA | 548.5 ± 98.2 bA | 1050.0 ± 230.8 aA | 420.4 ± 290.3 bA |
| V | | 46.96 ± 22.66 aA | 59.58 ± 3.42 aA | 26.88 ± 12.18 bA | 54.51 ± 12.20 aA | 58.90 ± 6.42 aA | 14.82 ± 21.20 bA |
| Zn | | 73.7 ± 39.9 aA | 88.53 ± 12.05 aA | 100.4 ± 32.7 aA | 96.45 ± 10.74 aA | 75.47 ± 10.68 aB | 29.7 ± 35.4 bB |
| β-glucosidase | | 64.51 ± 1.43 bA | 68.08 ± 2.03 aA | 55.38 ± 1.31 cA | 61.45 ± 1.10 bB | 66.60 ± 4.12 aA | 52.68 ± 0.93 cB |
| Alkaline phosphatase | | 52.92 ± 1.10 bA | 63.04 ± 1.45 aA | 53.52 ± 0.87 bA | 51.67 ± 0.56 cB | 63.00 ± 0.70 aA | 52.76 ± 0.52 aB |
| Acid phosphatase | mg p-nitrophenol g$^{-1}$ h$^{-1}$ | 55.35 ± 1.03 bA | 61.48 ± 1.65 aA | 52.60 ± 1.06 cA | 51.65 ± 0.83 bB | 62.19 ± 1.05 aA | 52.20 ± 0.24 aA |
| Arylsulfatase | | 52.49 ± 1.22 bA | 62.65 ± 1.33 aA | 52.87 ± 0.45 bA | 49.70 ± 0.43 cB | 59.36 ± 0.12 aB | 52.11 ± 0.14 aB |
| Urease | mg NH$_4^+$-N kg$^{-1}$ h$^{-1}$ | 112.16 ± 4.28 bB | 130.22 ± 3.03 aB | 102.47 ± 1.21 cB | 194.12 ± 0.61 bA | 232.93 ± 0.98 aA | 194.28 ± 0.57 bA |

WHC: water holding capacity, EC: electrolytic conductivity, CEC: cation exchange capacity, TOC: total organic carbon. Groups not sharing a letter are significantly different from each other (*p* > 0.05). Lower case letters indicate significant differences between soils of the same season, and upper-case letters between soils of different seasons. ND: not detected. Values are the mean of the results per indicator ± standard deviation (9 replicates).

The pH data ranged between 6.63 and 9.31, indicating that these soils were neutral and alkaline, with some within the optimal pH range—between 5.8 and 7.5—for agricultural soils [34]. The salinity levels and pH were associated with the presence and availability of salts at the sites, determining concentrations in the order of $SO_4^{2-} > Cl^- > HCO_3^- > CO_3^{2-}$ in both seasons. The presence of 16 elements was also observed in the soils—among them, high levels of Ca, Fe, Mg, Mn, and Li, characteristic of the parent rocks of a brackish area, which is suggestive of the area's geological origin [5,14].

Enzyme activities related to nutrient cycling showed a significant trend in S2 and S1 for the dry season and S2 and S3 for the rainy season in the five enzyme activities. The evaluation of biological indicators in soils is usually highly sensitive because they demonstrate the availability of nutrients and reflect the activity of microbial populations [29,35].

### 3.2. Development of the Soil Quality Index

Soil variables that showed significant differences between sites or seasons were included in the PCA and considered as members of the TDS. The first four principal components (PC) had eigenvalues >1.0 and variance >5% and together explained 80.39% of the variance of the original data (Figure 2). More than one highly weighted variable was considered for each PC and considered as a candidate for the MDS, distributed as follows: PC1 had four variables (TN, $CO_3^{2-}$, glucosidase, and V) explaining 36.96% of the variance; PC2 presented 19.08% of the variance constituted by three variables (CEC, TOC, and urease); and PC3 and PC4 were represented by one variable each—Li and Zn, explaining 12.72% and 11.61% of the variance, respectively.

The PCA technique has been widely used by soil-quality researchers for its ability to introduce less subjectivity in data selection, to help reduce bias and data redundancy, and to select the most representative indicators from a dataset [24,25,36,37]. However, this method requires an initial large dataset, more time for sampling and laboratory analyses, and more complex data interpretation [29]. On the other hand, the Spearman matrix

($r^2 > 0.6$) identified that all variables presented at least one significant correlation (Figure 3); therefore, we proceeded to the sum of the absolute values of the correlation coefficients of each variable. The NT, CEC, Li, and Zn variables (Table 2) were selected as best representing each component, and were designated as members of the MDS to constitute the SQI in the soils of the geothermal zone of "Los Negritos".

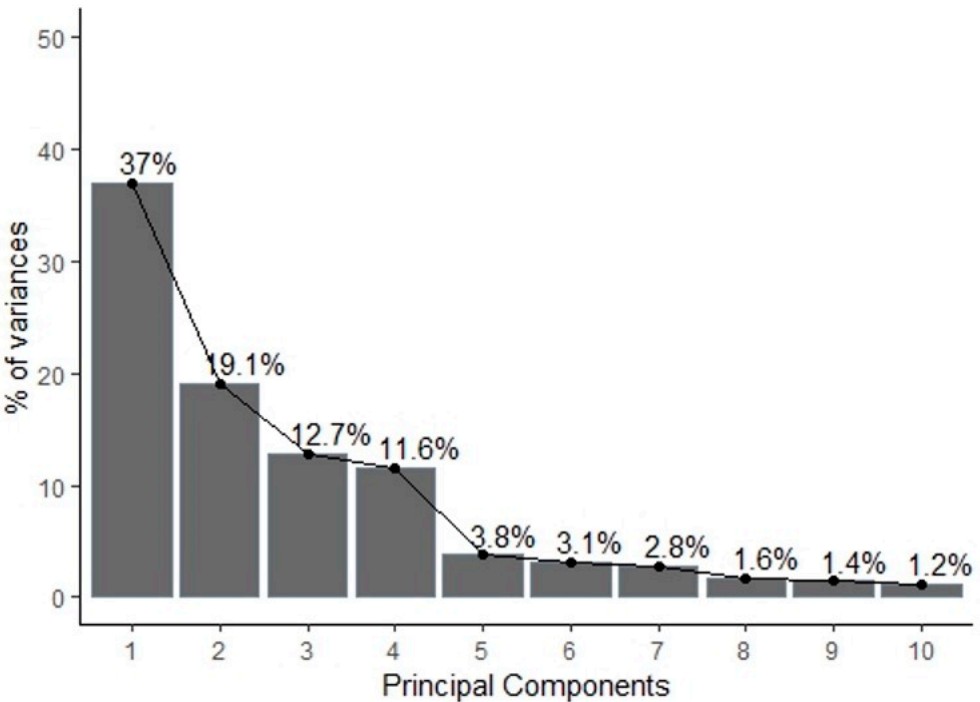

**Figure 2.** Percentage variance of TDS.

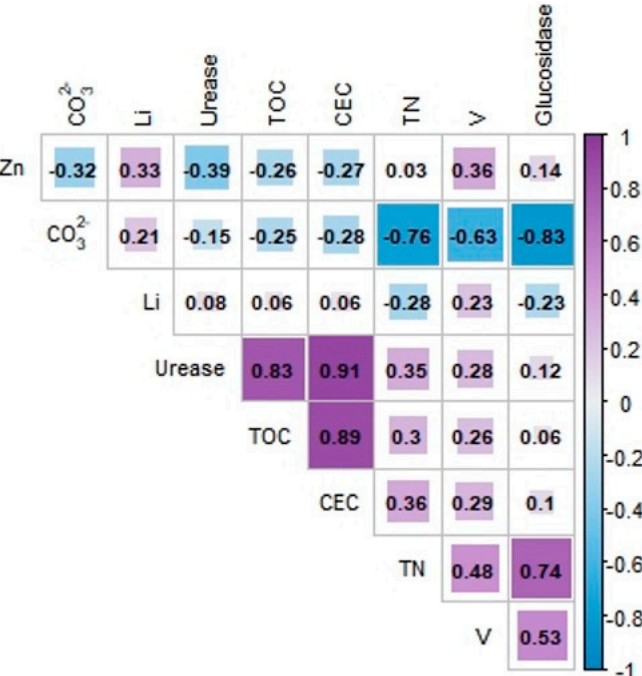

**Figure 3.** Spearman's correlation matrix. Indicators with positive correlation in purple, indicators with negative correlation in blue.

**Table 2.** Principal component analysis (PCA) output of the studied soil properties.

| Principal Component | PC1 | PC2 | PC3 | PC4 |
|---|---|---|---|---|
| Eigenvalue | 14.41 | 7.44 | 4.96 | 4.52 |
| Variance % | 36.96 | 19.08 | 12.72 | 11.61 |
| Cumulative % | 36.96 | 56.05 | 68.78 | 80.39 |
| CEC | 0.260 | **0.936** | 0.023 | −0.074 |
| TOC | 0.153 | 0.910 | 0.068 | −0.134 |
| NT | **0.897** | −0.033 | 0.264 | −0.151 |
| $CO_3^{2-}$ | −0.882 | 0.138 | −0.280 | −0.081 |
| Glucosidase | 0.866 | −0.212 | 0.129 | −0.220 |
| Urease | 0.226 | 0.921 | −0.077 | −0.235 |
| Li | −0.170 | 0.165 | **−0.641** | 0.515 |
| V | 0.906 | −0.029 | −0.091 | 0.289 |
| Zn | 0.440 | −0.212 | −0.339 | **0.717** |

CEC: cation exchange capacity, TOC: total organic carbon, and NT: total nitrogen. Bold numbers are the correlated parameters that contribute most to each CP, and indicators are considered as MDS.

Nitrogen—a critical macro-element in soil due to the plant biochemical processes in which it is involved, from root growth to maturation, including photosynthesis and nitrogen fixation [38]—was the most weighted indicator in this study. Nitrogen facilitates ecosystem balance and productivity due to N transformations driven by different microbial groups with different metabolic versatility and environmental tolerance [39]. Together with soil organic matter (SOM), nitrogen is considered a key component contributing to soil fertility [16]. The next indicator selected was the CEC, which represents the sum of exchangeable cations in the soil (K, Ca, Mg, and Na), and several available micronutrients [40]. CEC has been considered important for evaluating soil productivity in semi-arid ecosystems and for problems of high calcium content [14] because it shows the reserve of nutrients; a high level is associated with important levels of organic C in soils, essential for biological activity [7]. Lithium, another indicator chosen, is an element related to geothermal zones, brines, and magmatic and sedimentary rocks [41]. Natural Li concentrations depend on characteristics such as lithology, temperature, salinity, and water–rock interaction [42]; furthermore, Li has economic value in numerous industries, such as ceramics, glass, polymer production, and energy devices (batteries), which has increased its presence in agricultural soils [43]. It is an element whose role in the development of plants and animals—including humans—is unclear. Its toxicity has been reported at different concentrations [44]. In plants, Li influences physiology and biochemistry, reduces growth, and causes oxidative damage to the photosynthetic apparatus, metabolite composition, and nucleic acid and protein synthesis [45]. Zinc was the last designated indicator; it is an essential plant micronutrient involved in the synthesis of proteins, nucleic acids, and carbohydrates, as well as in the activation of enzymes and cell differentiation. Zinc accumulation in the soil depends on particles such as iron oxides and calcites that cause low availability for plant uptake [25]. When contamination is suggested, it is related to irrigated crops or soils adjacent to industrial areas [45].

The variables that comprised the MDS were transformed using (non-linear) scoring equations in terms of the property and its function in the soil [27,28]. Using Equations (1) and (2), a "more is better" curve was applied to the NT and CEC indicator, and "optimal" was given for Li and Zn. The transformed indicator scores and the values of the contribution of the individual indicators to the variance of their respective PC were then integrated into additive and weighted SQI using Equation (3), as follows:

$$\text{SQI}_w = \sum (\text{NT score} \times Si \text{ score}) + (\text{CEC score} \times Si \text{ score}) + (\text{Li score} \times Si \text{ score}) + (\text{Zn score} \times Si \text{ score}).$$

The sum of these values gave the soil quality indices for the three sites in both seasons. In the dry season, S3 had the highest soil quality—and this result was significant ($p \leq 0.05$)—whereas S1 and S2 had the highest soil quality in the rainy season (Figure 4). Furthermore, globally (during the annual cycle), no significant differences were observed between the

three sites (Figure 5). The opposite occurred between seasons, with the statistically highest-quality soils occurring in the rainy season ($p \leq 0.05$) (Figure 5). A better knowledge of soils is crucial to maintaining or increasing soil sustainability, identifying the most relevant soil attributes, and monitoring the changes generated by an event in each area [16,30]. Within soil quality observation and assessment studies, there are several methodologies for achieving this, ranging from multivariate geostatistical methods, factorial statistics, assessments based on crop growth and/or yields, visual assessment methods, expert opinion, and scoring and weighting methods for a set of indicators. The SQI formulated from obtaining MDS and non-linear scores—evidenced as a low-cost quantitative method due to the reduced number of indicators evaluated—provides the necessary information for decision making through the variability and sensitivity of indicators that represent the effects of changes in soil management and seasonality [14,25,27,29,46,47].

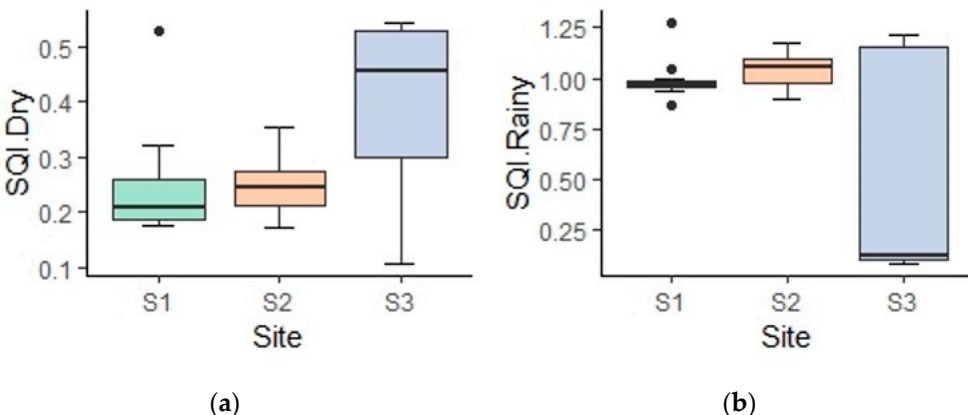

(**a**)                                          (**b**)

**Figure 4.** Box plots showing median soil quality index (SQI) levels between seasons: (**a**) dry season SQI and (**b**) rainy season SQI. Black dots indicate outliers.

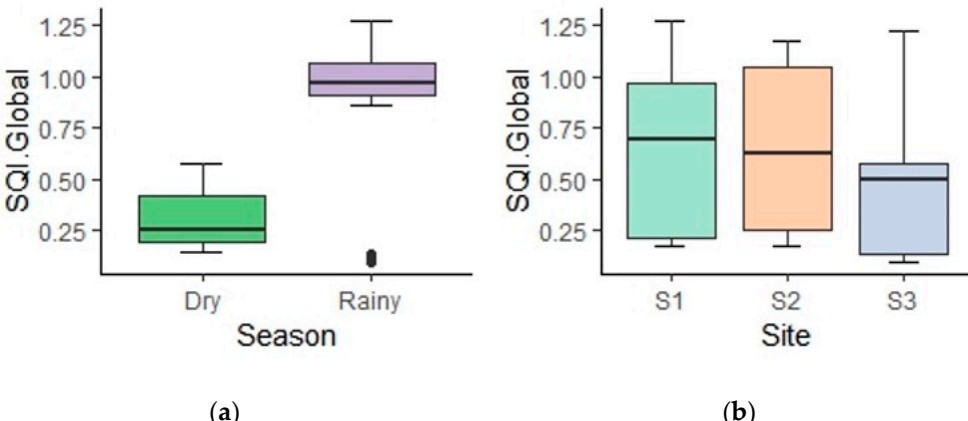

(**a**)                                          (**b**)

**Figure 5.** Box plots showing median global soil quality index (SQI) levels between season and sites: (**a**) global SQI season and (**b**) global SQI site. Black dots indicate outliers.

### 3.3. Soil Quality Grades

Five different levels of soil quality were established for this study (Table 3), for which the sites were assessed in the two seasons (Table 4) and globally (Tables 5 and 6). The global assessment of the sites was carried out considering the data from both stations without differentiating between them; similarly, the stations were assessed without differentiating between the sites. The soils were classified as "high". Between seasons, the dry season were classified as "low" and for the rainy season as "high".

**Table 3.** Soil quality classes and saline soil index values.

| Soil Quality | Very Low | Low | Moderate | High | Very High |
|---|---|---|---|---|---|
| Scale | <0.16 | 0.17–0.32 | 0.33–0.48 | 0.49–0.64 | >0.8 |
| Class | I | II | III | IV | V |

**Table 4.** Soil quality classes of saline sites between seasons.

| Season Site | Dry | | | Rainy | | |
|---|---|---|---|---|---|---|
| | 1 | 2 | 3 | 1 | 2 | 3 |
| SQIw | 0.26 bB | 0.24 bB | 0.43 aA | 0.99 aA | 1.04 aA | 0.56 bA |
| Soil Quality Class | Low | Low | Moderate | Very High | Very High | High |

Means that do not share a lower case letter are significantly different between sites in the same season, and the upper case letters denote significant difference between sites in different seasons. ($p \leq 0.05$).

**Table 5.** Global soil quality classes of saline sites.

| Site | 1 | 2 | 3 |
|---|---|---|---|
| Global SQIw | 0.62 a | 0.64 a | 0.49 a |
| Soil Quality Class | High | High | High |

Means that do not share a letter are significantly different ($p \leq 0.05$).

**Table 6.** Global soil quality classes between seasons.

| Season | Dry | Rainy |
|---|---|---|
| Global SQIw | 0.31 b | 0.86 a |
| Soil Quality Class | Low | High |

Means that do not share a letter are significantly different ($p \leq 0.05$).

The results showed a better performance of soil functions in the rainy season and in the sites with vegetation cover, because of the higher salinity concentration in the dry season and lower salinity concentration in the rainy season, closely related to the seasonal patterns of temperature, precipitation, pH, organic matter, the difference in agricultural practices, and the phenological cycle of plants.

In this study, water was an important factor in the soil, as it is a means of transport for the substrates in the hydrolysis processes, and in the control of microbial activity that determines mineralization rates, nutrient cycling, the maintenance of plant diversity, soil fertility, and ecosystem sustainability [4,13,48,49]. Similarly, crop residues on topsoil have been noted to promote microbial growth, decrease temperature, prevent erosion, and be a great source of material for mineralization [50]. On the other hand, low water content, high temperature, and little or no vegetation lead to high soil evapotranspiration, which causes the transport of large amounts of salt to the soil surface, affecting plant and bacterial communities and nutrient distribution, as well as altering the physical and chemical properties of the site and leading to a deterioration in soil functions [51].

The classification of the soil quality of the sites in this study was similar to those reported by several authors who generated their own classification in the search to quantify the soil quality of a particular area, such as the classification system of Cantú et al. [52], for evaluating soils with different agricultural uses and management; Karaca et al. [53], for grassland soils in semi-arid ecosystems; Mamehpour et al. [14], for evaluating urban cultivated soils in a semi-arid calcareous ecosystem; Santos-Francés et al. [47], for agricultural soils in a semi-arid ecosystem; and Sanchez-Navarro et al. [31], for soils in semi-arid Mediterranean regions.

## 4. Conclusions

Soil salinity is a latent threat to ecosystems and agricultural production by reducing plant growth and microbial functioning, so it is crucial to assess its interaction with other factors to

highlight the critical needs of a soil. Currently, there is no universal SQI that can be used in multiple natural and anthropogenic ecosystems, so targeted indexing strategies have been developed and implemented for specific environmental conditions around the world.

The applied methodology reduced the number of physical, chemical, and biological indicators analyzed from 39 to only 4, which allowed the establishment of a minimum set of data focused on indicating the quality of the soil in the study area. The soil quality classification proposed in this study during the annual monitoring evidenced the quality class of the sites: high, including two classifications for seasons: low and high. The analysis between seasons allowed us to propose a strategy to improve saline soils: the addition of organic materials such as plant residues to improve the nutrient content and the activity of microbial tolerance to saline stress. Likewise, the integrated quality index represented an effective tool to evaluate the impact of saline soil management practices and seasonality in an adequate and quantitative manner on soil functions with the use of selected MDS (NT, CIC, Li, and Zn). The selected indicators indicated a higher sensitivity in the chemical properties. However, the assessment of other biological properties that better reflect microbial activity and that can be used to relate abiotic soil properties in terms of biochemical transformations, as well as vegetation potential and performance under the same ecological conditions in this region, could be considered for practical, economical, and reliable results.

**Author Contributions:** Conceptualization, Y.B.-O., M.S.V.-M. and J.J.P.; Data curation, Y.B.-O. and M.S.V.-M.; Formal analysis, Y.B.-O.; Funding acquisition, M.S.V.-M. and J.J.P.; Investigation, Y.B.-O.; Methodology, Y.B.-O., M.S.V.-M. and M.O.F.-H.; Project administration, M.S.V.-M.; Resources, Y.B.-O.; Validation, Y.B.-O., M.S.V.-M. and J.J.P.; Visualization, Y.B.-O.; Writing—original draft, Y.B.-O., M.S.V.-M. and J.J.P.; Writing—review and editing, M.S.V.-M. and J.J.P. All authors have read and agreed to the published version of the manuscript.

**Funding:** This study was financially supported by the projects of Instituto Politécnico Nacional (IPN) (Nos. SIP20200229 and SIP 20210819) Prolongación Carpio y Plan de Ayala s/n, Col. Santo Tomás, Del. Miguel Hidalgo, C.P. 11340. Ciudad de México, Mexico, by grants from the Agencia Estatal de Investigación, AEI, Spain, (No. PID2021-125371OB-I00), and the Agencia Estatal Consejo Superior de Investigaciones Científicas, CSIC, Spain (No. COOPA20458).

**Data Availability Statement:** Data are available from corresponding authors.

**Acknowledgments:** Yanely Bahena-Osorio received grant-aided support from Consejo Nacional de Ciencia y Tecnología (CONACyT) and Beca de Estímulo Institucional de Formación de Investigadores-IPN (BEIFI). Marina Olivia Franco-Hernández, and María Soledad Vásquez-Murrieta received grant-aided support from Comisión de Operación y Fomento de Actividades Académicas-IPN (COFAA), Estímulos al Desempeño de los Investigadores-IPN (EDI), and Sistema Nacional de Investigadores-CONACyT (SNI). We thank Dioselina Álvarez-Bernal, Salvador Ochoa-Estrada, and Leonardo Yoguez-Alcantar (Centro Interdisciplinario de Investigación para el Desarrollo Integral Regional, Unidad Michoacán) for access and guidance in soil sampling, and Ciro Eliseo Márquez-Herrera (Facultad de Química UNAM, Mexico) for support in ICP-OES analysis.

**Conflicts of Interest:** The authors declare no conflict of interest.

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
