# Peer review of "Development of a Quality Index to Evaluate the Impact of Abiotic Stress in Saline Soils in the Geothermal Zone of Los Negritos, Michoacán, Mexico"

_agronomy, doi:10.3390/agronomy13061650_

Round 1

Reviewer 1 Report

The authors aimed to develop an integrated soil quality index for saline soils. In total, 39 indicators related to attributes of soil fertility and cyclic of main bio-elements were analyzed, and principal component analysis (PCA) and Spearman correlation matrix were applied to develop a soil quality index. The soil attributes were determined in two arable fields and one non cultivated site of high salinity. Soil parameters were established during rainy and dry seasons in two years. The studies showed that minimum data set for integrated quality index equation consisted of 4 indicators (N total, CEC, Li and Zn), and significantly higher SQI was found during rainy than dry season.

The manuscript is well written, and statistics analyzes properly applied. I have some remarks about methods and conclusions. A weak point of studies: there is no relation between soil quality index and yield, so we don’t know how effective the  index could be.

In my opinion, the manuscript needs some improvements before acceptance for publication.

Author Response

Answers to the reviewers

We are greatly thankful for all the comments and observations that helped us to improve our manuscript. We give our reply to your comments in the following section.

- Reviewer 1

  1. Is the soil compaction has no impact on arable land in the studied area?

A: The bulk density of the soil was not determined at the sites. However, the use of agricultural machinery during soil preparation is reported, as well as the use of means of transport to move the harvested products, which could influence the compaction of the sites.

  1. Soil management practices are not clearly described. Some information is given in lines 85-89, but what means that S1 “was ready for cultivation and S2 had sparse plant residue .. on the soil surface”. What make a difference between state of soils in both sites?

A: In the first season, S1 was ready for cultivation, i.e. medium deep furrows were made in the soil using agricultural machinery (tillage) to initiate an agricultural cycle; on the other hand, S2 had little plant residues of sugar cane (Saccharum officinarum L.) and weed growth on the soil surface, i.e. no tillage was carried out and no initial agricultural cycle was planned. This is now clarified in the manuscript.

  1. Can you precise how % of the area was covered by crop residue?

A: Approximately 80% of the S2 area was covered by crop residues.

  1. Can you explain why some rather stable soil properties (sand, silt, clay, total C) are so much differentiated during dry and rainy season?

A: TOC content persists for different periods of time and space, depending on environmental conditions, chemical structures, dynamics of deposition, decomposition, and transformation. Thus, the amounts of labile TOC can be higher and are more variable over time due to the action of the microbial community, depending on humidity and temperature.

On the other hand, the arrangement of the primary soil particles (sand, silt and clay) into aggregates periodically disintegrates and reforms. However, their stability is influenced by human factors, such as tillage, which destroys them by mechanically breaking them up and induces a reduction of infiltration and gas exchange. However, physical and chemical forces (van der Waals), physical forces (surface tension) and various cementing agents such as organic matter, exchange cations, plants, plant residues, metabolic products of soil macrofauna and water promote the structuring of these soils. The difference in clay content in S1 can be explained by these two events: tillage in the first season and the presence of crops in the second season.

  1. Can we really use the same notification (as S1) for soils that are differed so much in clay content?

A: The assignment of S1 is based on geographical location, and the purpose is to observe the changes in both seasons, the changes in clay content are explained in the previous point.

  1. Can you add an information about depth of sampling?

A: Sampling data, including depth, was carried out according to the criteria established by Guevara-Luna [17]. Soil samples were collected from nine points per plot, sampling the top layer of 15-25 cm after removing the top layer of 0-15 cm. This point is included in the main text.

  1. Did the variables were scaled before the application of PCA?

A: Scaling was performed using the prcomp() function and the argument scale=TRUE, which centres variables with means of zero and standard deviation of one.

  1. The authors use sometimes terms of “physicochemical”, “physic-chemical” or “physical, chemical”. It seems that use one term could be better (may be the latter one).

A: Thank you for your observation. It has been modified and “physical” and “chemical” was used.

  1. L187-188: “Statistical analyses (ANOVA, Tukey, PCA, Spearman correlation) were performed with MINITAB 17 and R 4.21 (www.r-project.org).” I have some doubts about a separation “Tukey test” as a separate statistical analysis. It is more like a help procedure to interpret ANOVA..

A: Your statement is correct, Tukey's method is used within ANOVA, it creates confidence intervals and allows discerning whether the results obtained are significantly different or not, contributing to a better interpretation. This part is corrected in the revised version.

  1. L310-312: “The results showed a better performance of soil functions in the rainy season and in the sites with vegetation cover, because of the higher salinity concentration in the dry season and lower salinity concentration in the rainy season, closely related to the seasonal patterns of temperature, precipitation, and the phenological cycle of plants.” However, if we look at values of EC (Table 1), the difference between EC in particular sites are not so big (S1 – 2.23 (dry) and 1.18 (rainy), S2 – 12.4 (dry) and 10.2 (rainy), S3 – 34.4 (dry) and 27.0 (rainy)). So, maybe there is other factor that affect a better performance of soil functions in rainy than dry season?

A: The EC values at the sites in both stations show statistically significant differences. Salinity concentration is an important factor limiting soil functions, however, as mentioned in L334-335, temperature, precipitation, pH, organic matter, the difference in agricultural practices and the phenological cycle of plants are other factors that contributed to the better performance of functions in the rainy season.

Reviewer 2 Report

The manuscript has improved the evaluation method of soil quality index. The authors take saline soils as the research object and use the method of integrated index equation and principal component analysis to calculate the soil quality index by screening the main contribution index values, which significantly reduces the amount of data and evaluation cost compared with the traditional evaluation method. The manuscript has sufficient data, a complete and logical structure, and objective and reasonable evaluation results, but there are still areas that need improvement and additional clarification, and specific suggestions for revision are as follows.

1. In the introduction section, there is a lack of introduction to the current status of research on soil quality indices. The author also mentions that existing research has not yet formed a consistent standard, but the author does not elaborate on previous research results and the shortcomings that exist, which is also the purpose of the author's research using the Integrated Quality Index (IQI).

2. In the section on soil collection, the authors refer to literature 17, but the title of the literature does not correspond to the actual one. It is recommended that the authors double-check the references.

3. It is suggested that the authors improve the clarity of Figure 1 and that the locations of S1 and S2 in the figure seem to be roads between cultivated fields.

4. In the calculation of SQI, the authors conducted a principal component analysis and assigned a weight value to each principal component.

5. In selecting the MDS indicators, the authors selected the four indicators with the largest contribution values from each of the four sets of principal component analysis results to form the MDS, but TN does not seem to be the value that contributes the most in PC1, V is the largest.

6. In the notes to Table 2, the explanatory note for TN is misstated and should not be NT.

7. Table 5 shows that saline soil grades are high globally. Whether this result corresponds to the actual situation is not explained by the authors, who only give a brief description of the reasons for this result.

8. The title of this manuscript is an assessment of the quality of saline soils in the geotropics. What are the characteristics of the geotropics? Why did the author choose the geotropic zone? Please explain this in the introduction.

9. The Conclusion is very general. Some recommendations based on the findings of the study need to be included to make the study more meaningful.

Author Response

We are greatly thankful for all the comments and observations that helped us to improve our manuscript. We give our reply to your comments in the following section.

- Reviewer 2

  1. In the introduction section, there is a lack of introduction to the current status of research on soil quality indices. The author also mentions that existing research has not yet formed a consistent standard, but the author does not elaborate on previous research results and the shortcomings that exist, which is also the purpose of the author's research using the Integrated Quality Index (IQI).

A: Thank you for your observation. The introduction section has been improved taking your comments into considetation.

The main disadvantage is the unequivocal interpretation or the lack of reference values, which affects the subjectivity of the evaluated indicators, which is why it is important to clearly define the objectives of the study [12]. However, the Integrated Quality Index (IQI) is the most widely used index, because it proves to be a flexible, effective, and easy quantification tool for assessing the quality of a given soil or region. Also it reduces measurement costs by reducing the number of indicators used and it avoids collinearity [12,14,16]. Some studies have assessed soil quality using MDS to calculate the IQI. For example, Yuan et al. [8] assessed 12 soil properties and established the SQI using the parameters such as soil organic carbon (SOC), microbial biomass carbon (MBC), total potassium (TK), oxidation-reduction potential (Eh) and Mn (II) in soils with aquaculture activities. Mamehpour et al. [14] determined 24 variables and as a result, EC, OC, SAR, CEC, bioavailable Fe, total Cd and Pb were selected as MDS to evaluate soils in semi-arid calcareous ecosystems and Liu et al. [16], based on 26 parameters, established an MDS with soil organic matter (SOM), total nitrogen (TN), pH, dehydrogenase and arbuscular mycorrhiza for IQI in agricultural soils.

  1. In the section on soil collection, the authors refer to literature 17, but the title of the literature does not correspond to the actual one. It is recommended that the authors double-check the references.

A: The soil collection refers to literature [17] Guevara-Luna et al. We quote: "Each of the three sampling areas was divided into three plots, and soil samples were taken from nine points per plot sampling the top layer of 15-25 cm after removing the top layer of 0-15 cm. Subsequently, soil samples taken from the nine points in each plot were mixed to obtain composite soil samples (500 g, n=9)".

  1. It is suggested that the authors improve the clarity of Figure 1 and that the locations of S1 and S2 in the figure seem to be roads between cultivated fields.

A: The clarity of Figure 1 has been improved. Obviously, the locations were not roads, but agricultural lands.

  1. In the calculation of SQI, the authors conducted a principal component analysis and assigned a weight value to each principal component. Yes, we did, following standard methods, as indicated in the M&M section of the manuscript.

  1. In selecting the MDS indicators, the authors selected the four indicators with the largest contribution values from each of the four sets of principal component analysis results to form the MDS, but TN does not seem to be the value that contributes the most in PC1, V is the largest.

A: The data in table 2 represents the values of the contribution of the variables to the composition of each PC. However, all variables were redundant so the absolute values of the correlation coefficients of each variable were added (data not shown) and the variable with the highest correlation sum was selected to best represent the PC and form the MDS, assigning those parameters in bold.

  1. In the notes to Table 2, the explanatory note for TN is misstated and should not be NT.

A: Thank you for your observation. The NT was changed to TN.

  1. Table 5 shows that saline soil grades are high globally. Whether this result corresponds to the actual situation is not explained by the authors, who only give a brief description of the reasons for this result.

A: Table 5 shows the soil quality classes of the saline sites globally, the assessment of the sites was carried out considering the data from both stations without differentiating between them, similarly, the stations were assessed without differentiating between the sites. This has been clarified in the manuscript.

  1. The title of this manuscript is an assessment of the quality of saline soils in the geotropics. What are the characteristics of the geotropics? Why did the author choose the geotropic zone? Please explain this in the introduction.

A: Geothermal sites were described by Guevara-Luna et al. [17]. I quote "Los Negritos is a geothermal field located within the eastern zone of Lake Chapala, Michoacán, Mexico, on the limit of the Trans-Mexican Volcanic Belt. Hydrothermal activity has been reported in this area and is associated with the presence of mud volcanoes with temperatures between 48 and 94 â—¦C at the surface”. this has been clarified in the manuscript.

  1. The Conclusion is very general. Some recommendations based on the findings of the study need to be included to make the study more meaningful.

A: Thank you for your observation. The conclusion has been improved.

Soil salinity is a latent threat to ecosystems and agricultural production by reducing plant growth and microbial functioning, so it is crucial to assess its interaction with other factors to highlight the critical needs of a soil. Currently, there is no universal SQI that can be used in multiple natural and anthropogenic ecosystems, so targeted indexing strategies have been developed and implemented for specific environmental conditions around the world.

The applied methodology reduced the number of physical, chemical, and biological indicators analysed from 39 to only 4, which allowed establishing a minimum set of data focused on indicating the quality of the soil in the study area. The soil quality classification proposed in this study during the annual monitoring evidences a quality class of the sites: high, including two classifications for seasons: low and high. The analysis between seasons allowed us to propose a strategy to improve saline soils; the addition of organic materials, such as plant residues to improve the nutrient content and the activity of microbial tolerance to saline stress. Likewise, the integrated quality index represented an effective tool to evaluate the impact of saline soil management practices and seasonality in an adequate and quantitative manner on soil functions with the use of selected MDS (NT, CIC, Li, and Zn). The selected indicators indicate a higher sensitivity in the chemical properties. However, the assessment of other biological properties that better reflect microbial activity and that can be used to relate abiotic soil properties in terms of biochemical transformations as well as vegetation potential and performance under the same ecological conditions in this region could be considered for practical, economical and reliable results. Your comment helped us improve the Conclusions sections.
